# Me-Better Drug Design Based on Nevirapine and Mechanism of Molecular Interactions with Y188C Mutant HIV-1 Reverse Transcriptase

**DOI:** 10.3390/molecules27217348

**Published:** 2022-10-29

**Authors:** Yan Wang, Aidong Wang, Jianhua Wang, Xiaoran Wu, Yijie Sun, Yan Wu

**Affiliations:** 1Key Laboratory of Drug Design, College of Chemistry and Chemical Engineering, Huangshan University, Huangshan 245061, China; 2Key Laboratory of Biorheological Science and Technology, Ministry of Education, Chongqing University, Chongqing 400044, China

**Keywords:** nevirapine, fragment growth, Y188C mutation HIV-1 reverse transcriptase, HIV-1, molecular docking

## Abstract

In this paper, the Y188C mutant HIV-1 reverse transcriptase (Y188CM-RT) target protein was constructed by homology modeling, and new ligands based on nevirapine (NVP) skeleton were designed by means of fragment growth. The binding activity of new ligands to Y188CM-RT was evaluated by structural analysis, ADMET prediction, molecular docking, energy calculation and molecular dynamics. Results show that 10 new ligands had good absorbability, and their binding energies to Y188CM-RT were significantly higher than those of wild-type HIV-1 reverse transcriptase(wt). The binding mode explained that fragment growth contributed to larger ligands, leading to improved suitability at the docking pocket. In the way of fragment growth, the larger side chain with extensive contact at terminal is obviously better than substituted benzene ring. The enhancement of docking activity is mainly due to the new fragments such as alkyl chains and rings with amino groups at NVP terminal, resulting in a large increase in hydrophobic bonding and the new addition of hydrogen bonding or salt bonding. This study is expected to provide reference for the research on non-nucleoside reverse transcriptase inhibitors resistance and AIDS treatment.

## 1. Introduction

It is widely known that AIDS is a worldwide medical problem that needs to be overcome urgently [1,2]. It is a malignant infectious disease caused by the HIV virus infection of T cells, resulting in the destruction of immune function and opportunistic tumor growth [3,4]. HIV is an RNA virus with two subtypes, HIV-1 and HIV-2, with the former being mainly studied by scientific researchers due to its characteristics of strong infectivity, high mortality and global popularity [5,6]. HIV-1 reverse transcriptase (RT) is a multifunctional and essential enzyme in the life cycle of HIV-1 which has great activities with RNA-dependent DNA polymerase, DNA-dependent DNA polymerase and ribonuclease [7,8,9]. Today, two kinds of nucleoside reverse transcriptase inhibitors and non-nucleoside reverse transcriptase inhibitors (NNRTI) are approved for the treatment of HIV-1 infection [6,10].

As a commonly used NNRTI for HIV-1 treatment, NVP can combine with HIV-1 RT and destroy the catalytic site of RT, thus blocking the activity of DNA polymerase and HIV replication [11,12]. However, continuous usage of nevirapine (NVP) induces drug-resistant mutations in HIV-1 virus and further failure of NNRTI treatment [13,14,15]. Y188C mutant HIV-1 reverse transcriptase (Y188CM-RT) is an important mutation that leads to NVP resistance in vivo [16,17,18]. In order to overcome NVP resistance, it is urgent to optimize its structure and improve inhibition activity against HIV-1 RT mutants.

Molecular simulation performs well as a non-destructive technique with low cost and simple operation which has increasingly wider application in molecular science and drug design [19,20,21,22]. In this study, the 3D Y188CM-RT model constructed by homologous modeling was selected as the target receptor, and new ligands based on the NVP skeleton were designed by fragment growth. Then, structural analysis, ADMET prediction, molecular docking and energy calculations were carried out to obtain the binding activity of new ligands with Y188CM-RT. Finally, the interaction mode between the new NVP-based ligands and Y188CM-RT was accurately explained by molecular docking and molecular dynamics with vivid visualization. This study is expected to provide references for anti-HIV-1 drug development and disease treatment.

## 2. Experimental

All ligand molecules applied in this paper were drawn using BIOVIA Draw 2016. All molecular docking simulations, data analysis and binding energy calculations were completed using software Discovery Studio Client v16.1.0.15350 (DS 2016) [23] and visualized with Pymol 2.3.0 (New York, NY, USA).

### 2.1. Homology Modeling, Model Evaluation and Target-Site Determination

The Protein Data Bank (PDB) was accessed to retrieve the amino acid sequence of Y188CM-RT (PDB ID, 1JLF). Herein, Y188L mutant HIV-1 Reverse Transcriptase (PDB ID, 2ynf) was selected as a template and a Swiss model was applied for homology modeling, achieving 3D structure model of Y188CM-RT. Subsequently, the evaluation of the Y188CM-RT model was performed using DS 2016. Finally, the reasonable binding site for ligands was determined after energy optimization [16,24]. 

### 2.2. Fragment Growth and ADMET Prediction 

After hydrogenation and energy optimization of NVP, fragment growth was generated by protocol of De Novo Evolution based on receptor–ligand interaction [25,26]. Referring to publications [27,28,29], these new ligands were added fragments without changing binding site inside Y188CM-RT, named Lig 1 to Lig 10, respectively. 

ADMET prediction was performed using DS 2016 to evaluate absorption and toxicity of new NVP-based ligands. Here, aqueous solubility, blood–brain barrier (BBB) penetration level, cytochrome P450 2D6 enzyme inhibition (CYP 2D6), and human intestinal absorption (HIA) level were estimated. Meanwhile, features of the hepatotoxicity, mutagenicity, degradability and rat oral LD*_50_* were predicted to screen Y188CM-RT inhibitors. 

### 2.3. Molecular Docking

Optimized NVP was semi-flexibly docked with Y188CM-RT using the CDOCKER model in DS 2016, with the parameters setting to default values. The clustering radius and resolution were set to 0.5 Å and 2.5Å, respectively. The energy function (-CDOCKER interaction energy) was selected as the evaluation value, in which the effects of non-bonding interactions were considered to be including hydrogen bonding, hydrophobic bonding, van der Waals force and electrostatic interactions. Obviously, the higher energy indicates a more stable complex and better associativity. The docking conformation with highest docking energy was the optimum, which was visualized by DS 2016 and Pymol.

The new ligands, Lig 1 to Lig 10, were completed in the same fashion as above. 

### 2.4. Molecular Dynamics Simulation 

Molecular dynamics simulation can verify the binding affinity between Y188CM-RT and NVP-based ligands [29,30]. The process was implemented by DS 2016 as follows: (1) we deleted redundant peptides, prepared a structure and applied a CHARm force field. (2) As shown in Figure 1, the complexes were placed into a cubic box filled with an aqueous solution (19,428 water molecules), while the whole system charge was balanced with 51 Na^+^ (purple sphere) and 55 Cl^−^ (green sphere). (3) Molecular dynamics simulation was completed by Standard Dynamics Cascade function, including energy optimization (default parameters), heating (20 ps), equilibration (300 ps) and production (400 ps). (4) After molecular dynamics simulation, results were analyzed using the Analyze Trajectory function. In addition, the averaged MM/PBSA protein–ligand binding free energy was calculated through MD trajectory. 

## 3. Results and Discussion 

### 3.1. Homology Modeling and Evaluation

The crystal structure of the Y188L mutant HIV-1 reverse transcriptase displayed an 91%-identified sequence with Y188CM-RT, and was selected as a template homology modeling. Then, 3D model of Y188CM-RT was achieved using Swiss model Figure 2a). Besides, the stereochemical structure of the Y188CM-RT model was detected, and its Ramachandran Plot was obtained by evaluating stereochemical stability of both main and side chains (Figure 2b). Results show that 99.40% amino acids are in the conformationally allowed regions and most are in the favored allowed regions, indicating the rationality of the Y188CM-RT model.

### 3.2. Binding Analysis of NVP-Y188CM-RT

Based on energy optimal conformation, analysis of an NVP-Y188CM-RT docking complex was performed, according to the specification of the publications [31,32,33]. The -CDOCKER interaction energy was 38.89 kcal/moL. Figure 3 presents the binding mode of the NVP-Y188CM-RT docking system. Figure 3a shows the active site on Y188CM-RT, and Figure 3b illustrates the docking pocket with NVP inside. Through amino acid sequence matching and Figure 3e, it is found that NVP inserted into the hydrophobic central cavity of Y188CM-RT and interacted with surrounding amino acid residues. Considering the 2D schematic diagram (Figure 3d) and certain amino acid residues (Figure 3c), Trp 229, Cys188, Leu 234, Tyr 181, Val 79, Val 105, Leu 100 and Lys 101 were involved in the binding process between NVP and Y188CM-RT. Specifically, there are 12 favorable bonds, including 11 hydrophobic bonds and 1 hydrogen bond (Lys 101, 2.54 Å). The hydrophobic bonds include 9 alky/π-alky (Leu 100, 4.36 Å; Val 106, 4.69 and 4.09 Å; Val 179, 5.38 Å; Leu 234, 5.50 Å; Cys 188, 5.05 and 4.49 Å; Trp 229, 4.33 Å), π–π stacked (Tyr 181, 5.04 Å) and π–σ (Leu 100, 2.78 Å) interactions. The results indicate that hydrophobic interaction mainly contributed to the stability of the NVP-Y188CM-RT complex.

### 3.3. Structural Analysis 

After treatment with De Novo Evolution, new NVP-based ligands were designed by means of fragment growth based on the NVP structure (shown in Figure 4). The structure characteristics of NVP are three carbon nitrogen heterocycles and one ternary carbon ring, while the main binding force with Y188CM-RT is driven by hydrophobic interactions. Therefore, two ways were promoted to effectively improve the binding affinity between NVP and Y188CM-RT. One method was to enhance hydrophobic interactions by adding hydrophobic carbon ring, and the other was to add new hydrogen bonds or salt bonds by adding N and O atoms.

Fragment growth was performed to fill the binding site, based on the residue properties and cavity size of receptor Y188CM-RT [34,35]. Herein, Figure 5 presents the fragment growth of Lig 1 as a typical example. It can be considered to introduce groups into P_1_ and P_2_ because of the large cavities at the positions. The cavity at P_1_ is smaller than that at P_2_ and is surrounded by the amino residues Trp and Val. Therefore, looping and introducing double bonds at P_1_ increased the hydrophobic interactions with Trp and Val, whereas the cavity at P_2_ is much larger than P*_1_* and surrounded by amino residues Glu, Lys and Val. The imidazole group was introduced into the terminal, while it was found that the imidazole was too small to fill the cavity and a linker (methyl) was used as a connection. After connection, the imidazole group was close to the nearby amino acids, resulting in the formation of salt bridge and hydrophobic interactions. Theoretically, fragment growth would not change the interactions between original ligand and receptor, but only increase or strengthen some interactions.

### 3.4. ADMET Prediction

Pharmacokinetic parameters and toxicity are important evaluation indicators reflecting applicability and feasibility, as well as solubility, permeability, bioavailability and distribution. The predicted values of ADMET parameters of NVP and derivatives are shown in Table 1. Acceptable range of some parameters are aqueous solubility ≥ 2, CYP 2D6 < 0, hepatotoxicity < 0, and HIA level ≤ 1. 

All ligands have memorable aqueous solubility and intestinal absorption. Moreover, they are free of cytochrome inhibition, hepatotoxicity, mutagenicity and degradation. Predicted values of rat oral LD_50_ are within expected ranges. Evidently, the predicted BBB penetration level of ligands are equal or higher than NVP, except Lig 8. This indicates a low BBB permeability and low destructiveness of environmental stability in brain tissue. The above shows that these 10 NVP-based ligands have good pharmaceutical properties and excellent bioavailability. 

### 3.5. Molecular Docking Studies

HIV-1 RT is an important target for developing AIDS drugs. NVP can non-competitively bind to HIV-1 RT, blocking the binding events between substrates and RT [36]. Theoretically, the better the affinity of drugs to RT, the higher the inhibition against the HIV-1 virus. Molecular docking was performed for binding evaluation between NVP and RT, including wt and Y188CM-RT. A total of 10 new ligands with higher scores are screened (Table 2), and herein wt was employed as a comparison with Y188CM-RT.

Through binding energies, it was possible to observe that several results were delineated here: (i) For NVP, the binding energy with wt (40.9570 kcal/mol) is higher than Y188CM-RT (38.8888 kcal/mol), illustrating the resistance of the Y188CM-RT virus against NVP. (ii) For NVP-based ligands, binding energies with Y188CM-RT are commonly higher than wt, revealing better inhibitory activities against Y188CM-RT. (iii) In comparison between ligands, the binding energies targeting the Y188CM-RT of 10 derivatives are higher than NVP itself. Among the 10 ligands, the 3 ligands with the highest binding energies are Lig 1 (52.3722 kcal/mol), Lig 9 (48.2571 kcal/mol) and Lig 10 (48.7245 kcal/mol), respectively.

To quantify the differences in binding energies with Y188CM-RT and wt, -CDOCKER interaction energies of NVP-based ligands were assessed by paired *t* test (Table 3). Results show statistical differences (*p* < 0.05) between Y188CM-RT and wt. Specifically, the mean value of Y188CM-RT (46.1941 kcal/mol) is significantly higher than wt (35.6550 kcal/mol). The above also shows the feasibility of drug optimization by fragment growth to overcome the drug resistance.

Y188CM is a missense mutation, with Tyr substituted by Cys at codon 188 [16,23]. After mutation, the pocket of RT combined with NNTRIs changed, and the internal space increased significantly [37,38]. Thence, the ligands with a larger size were suitably accommodated into the docking pocket. NVP-based ligands with fragment growth hold larger MW and structural size, demonstrating consistency with docking pockets inside Y188CM-RT. In addition, NVP is more dependent on ring-stacking interactions due to its multi-ring skeleton, so as to lose more binding energy with the mutant. For NVP, increasing the side chain with extensive contact at the end is significantly better than the substitute benzene ring. Therefore, Lig 1, 5, 7, 8, 9 and 10 have higher docking scores than other ligands, which is attributed to the addition of longer chains on the tail ternary ring.

In order to deeply analyze the binding interactions between new NVP-based ligands and active amino acid residues, the interaction modes of the docking complexes were clearly visualized. Figure 6 shows the docking mode of Lig 1, Lig 9 and Lig 10 as examples. Herein, Figure 6a–c are the 2D schematic interaction diagrams, presenting active residues as well as interaction type (including distance). Figure 6d exhibits the conformations of ligands (Lig 1, Lig 9, Lig 10 and NVP) in the binding pocket as tangential mode; these share an overlapping presentation due to the structural similarity among the four ligands. Thereby, the panorama of the active pocket with ligands inside is shown in Figure 6e, so as to promote the identification and visualization for different ligands. Figure 6f shows the comparison of active docking structures of Lig 9 and Lig 10 with NVP, while Figure 6g shows Lig 1. By associating Figure 6d–f, high similarities can be found in the docking conformations of Lig 9, Lig 10 and NVP, which is attributed to the similar derivations from the NVP molecular skeleton. Accordingly, the generated new fragments are mainly located at the edge area of the docking pocket, which neither significantly increases the steric hindrance of the ligand structure itself, nor affects the interaction between NVP’s own structural group, or atom and amino acid residues. Therefore, there are no significant changes in Y188CM-Lig 9 and Y188CM-Lig 10 complexes compared with Y188CM-NVP. The generated pyridine ring at P_1_ has no impact on their binding affinities. The newly added chain merely increases 1 hydrogen bond, on the basis of retaining other interactions. 

Whereas, the docking conformation of Lig 1 is obviously different from NVP through Figure 6e,g. It presents a conspicuous deformation with a certain angle torsion, changing the distances between atoms on Lig 1 and active residues. Inevitably, the interactions between Y188CM-RT and Lig 1 change accordingly. It is inferred that Lig 1 has the largest generated fragment, the longest chain and most atoms among all ligands. Taking Lig 1 as an example, the effect of fragment growth on binding interactions was interpreted in detail. Overall, Lig 1 retains original interactions between NVP and Y188CM-RT, and the newly added groups promote interactions with amino acid residues farther away. Obviously, the docking system newly increased 4 hydrophobic bonds (Lys 103, 4.49 Å; Val 108, 5.04 Å; Phe 227, 4.83 Å; Pro 236, 5.24 Å) and shortened the distance between the hydrogen donor on the ligand and the residue Lys 101 from 2.72 Å to 1.88 Å, resulting in the strengthening of the hydrogen bonding (Lys 101, 1.88 Å). Moreover, a salt bond (Glu 138, 2.40 Å) was newly generated due to the existence of amino groups on 1,3-imidazole. Essentially, adding new fragments at the NVP terminal can effectively improve its binding energy with Y188CM-RT by greatly increasing the hydrophobic bond and adding hydrogen bond or salt bond, which is consistent with the fragment growth theory. In addition, this effect is more obvious when the growth fragment is large enough.

### 3.6. Molecular Dynamics Simulation

Based on the above, the docking complexes Y188CM–ligands (NVP, Lig 1, Lig 9 and Lig 10) were selected for molecular dynamics simulation to explore their binding affinity. Root-mean-square deviations (RMSD) and residue root-mean-square fluctuations (RMSF) of Y188CM-ligand complexes were recorded and shown in Figure 7, which was compared with the first frame of docking complexes. It was observed that RMSD values gradually increased, but finally stabilized around specific values. The flexibility of amino acid residues in complexes was evaluated by estimating RMSF from MD trajectory. Figure 7b was unable to distinguish the difference in the flexibility that arises in response to different ligands binding to the same protein.

The averaged MM/PBSA Y188CM ligand binding free energies were calculated to estimate stability, and the results are displayed in Table 4. In general, the binding energy values of the three NVP-based ligands are slightly higher than those of NVP. Electrostatic energy performed a major role as the component of binding energies. Meanwhile, Van der Waals energy and polar solvation energy facilitated the protein–ligand binding, although the effect was weak. However, the surface solvation energy was unfavorable. 

## 4. Conclusions

In this study, new NVP-based ligands were designed by a fragment growth method, and their binding affinities with Y188CM-RT were studied. Fragment growth based on receptor–ligand binding sites is an effective method for structural optimization, which increases affinities with Y188CM-RT by filling the cavity with fragments or linkers. A total of 10 NVP-based ligands have significantly higher affinities with Y188CM-RT than wt, indicating the potential applicability to overcome NNTRI resistances. Besides, the binding mode demonstrated that adding large side chains with extensive contact at the terminal is an efficient way, including of nitrogen-containing alkyl chains or nitrogen-containing ring structures. On the basis of maintaining the interactions between NVP and Y188CM-RT binding complexes additionally added hydrogen bonds or salt bonds. Additionally, the larger the new fragment at the NVP’s terminal, the stronger binding affinity of the ligands with Y188CM-RT. Furthermore, molecular dynamics results showed the good stability of Y188CM-ligand complexes. Overall, this study shows that the NVP-based ligands with fragment-growth have stronger binding affinity with Y188CM-RT, which can provide references for NNTRIs development and AIDS treatment.

## Figures and Tables

**Figure 1 molecules-27-07348-f001:**
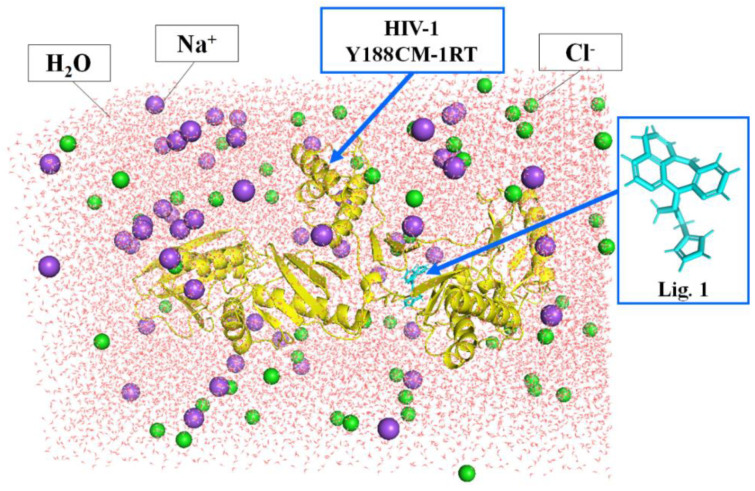
Docking complexes placed in a cubic box filled with water. Y188CM−RT and Lig 1 are shown as cartoon and stick models, respectively.

**Figure 2 molecules-27-07348-f002:**
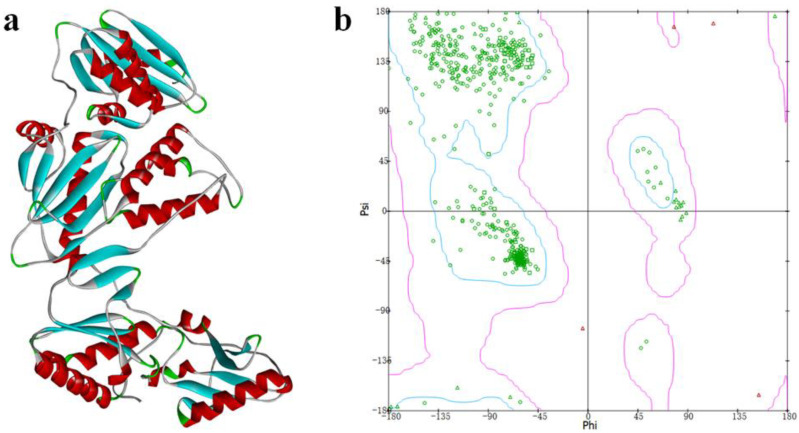
Y188CM-RT model (**a**) and Ramachandran Plot (**b**).

**Figure 3 molecules-27-07348-f003:**
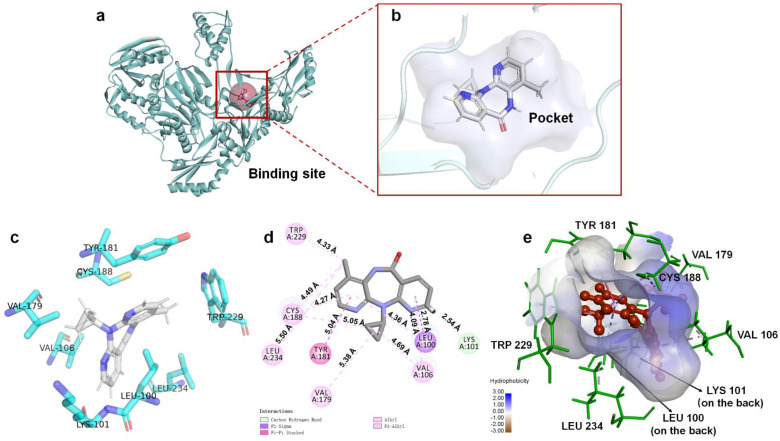
The binding mode between ligand NVP and receptor Y188CM−RT. (**a**) Spatial structure of Y188CM−RT (green) with the active binding site (red). The protein is presented as a cartoon model. (**b**) Docking pocket of Y188CM−RT with NVP inside. NVP is represented as a ball−stick model. C, N, O and H atoms are in gray, blue, red and white, respectively. (**c**) Active amino acid residues in the docking center. (**d**) 2D schematic interaction diagram between NVP and Y188CM−RT. (**e**) 3D docking mode between NVP and Y188CM-RT with surface of hydrophobic potential.

**Figure 4 molecules-27-07348-f004:**
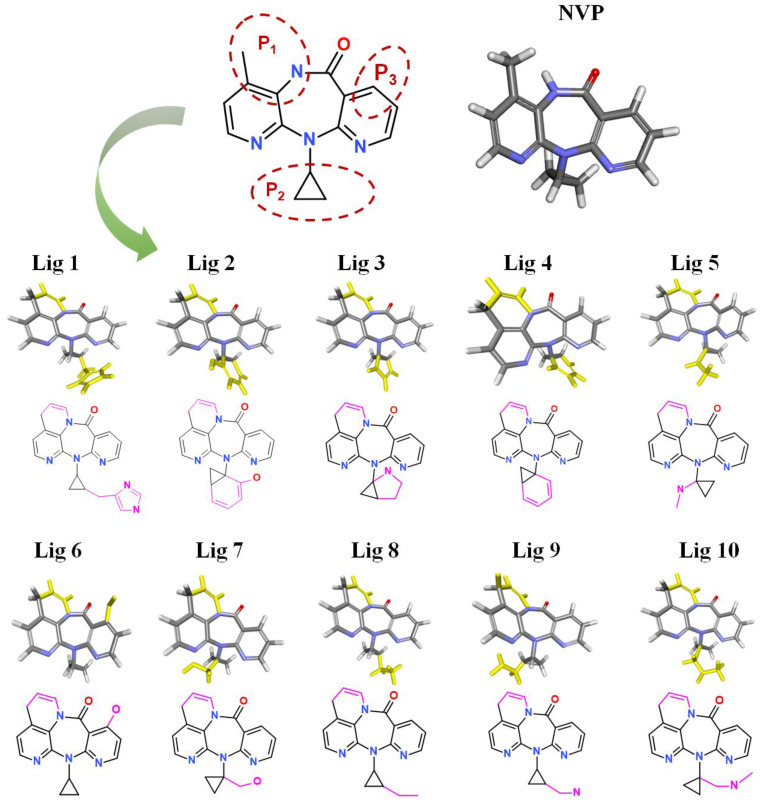
New ligands generated from NVP by means of fragment growth. Three positions to generate new fragments (red dotted line) and new fragments with purple-red marks.

**Figure 5 molecules-27-07348-f005:**
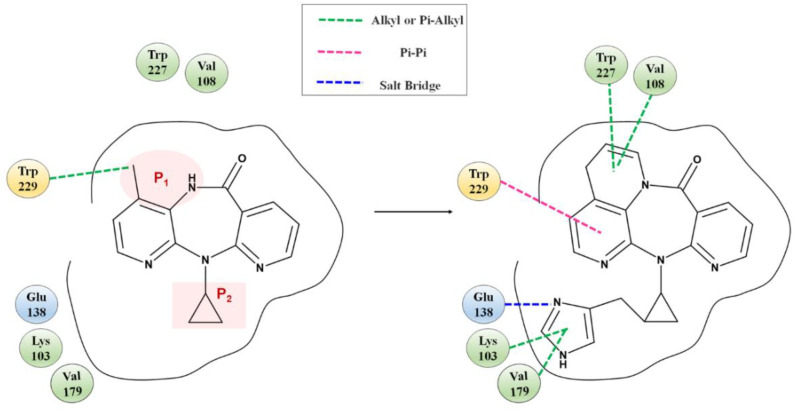
Fragment growth based on receptor–ligand binding site, taking Lig 1 as an example.

**Figure 6 molecules-27-07348-f006:**
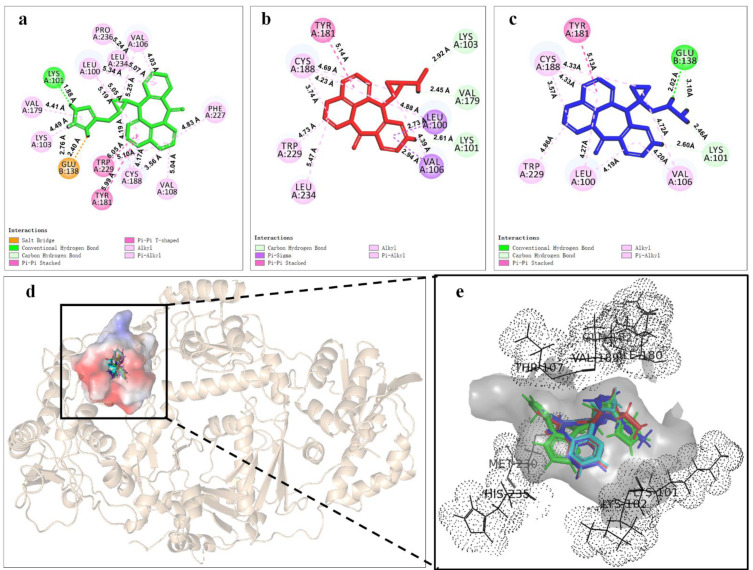
The binding mode between ligands (Lig 1, Lig 9 and Lig 10) and Y188CM-RT. (**a**–**c**) 2D schematic interaction diagram between ligands (Lig 1, Lig 9 and Lig 10, respectively) and Y188CM-RT. (**d**) The conformations of ligands in the binding pocket as tangential mode. (**e**) The docking pocket with the ligands inside. Ligands are shown as in the stick model. Lig 1, Lig 9, Lig 10 and NVP are in green, red, blue and cyan, respectively. (**f**,**g**) Comparison of active structures of Lig 9/10 and Lig 1 with NVP.

**Figure 7 molecules-27-07348-f007:**
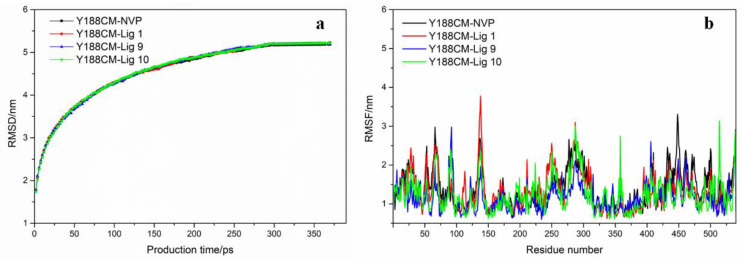
RMSD graph (**a**) and RMSF (**b**) of Y188CM–ligand complex graphs are shown for complexes with NVP (black), Lig 1 (red), Lig 9 (blue) and Lig 10 (green).

**Table 1 molecules-27-07348-t001:** Predicted values of ADMET parameters of NVP and derivatives.

Entry	Aqueous Solubility	BBB Penetration Level	CYP 2D6	Hepatotoxicity	HIA Level	TOPKAT Ames Prediction	TOPKAT Aerobic Biodegradability	TOPKAT Rat Oral LD_50_
NVP	2	2	−22.2017	−1.29011	0	Non-Mutagen	Non-Degradable	1.30686
Lig. 1	2	3	−5.4824	−1.58513	0	Non-Mutagen	Non-Degradable	1.21532
Lig. 2	2	3	−10.559	−1.27281	0	Non-Mutagen	Non-Degradable	0.13506
Lig. 3	3	3	−10.2184	−0.44516	0	Non-Mutagen	Non-Degradable	0.71373
Lig. 4	2	2	−8.29288	−1.20287	0	Non-Mutagen	Non-Degradable	0.32569
Lig. 5	3	3	−8.18333	−1.91685	0	Non-Mutagen	Non-Degradable	0.88734
Lig. 6	3	3	-8.99978	−0.95539	0	Non-Mutagen	Non-Degradable	0.07502
Lig. 7	3	3	−9.23876	−1.77251	0	Non-Mutagen	Non-Degradable	0.64649
Lig. 8	2	1	−7.08521	−1.85011	0	Non-Mutagen	Non-Degradable	0.98136
Lig. 9	3	3	−9.9088	−2.42656	0	Non-Mutagen	Non-Degradable	0.36748
Lig. 10	3	3	−6.8001	−2.66234	0	Non-Mutagen	Non-Degradable	0.37014

LD_50_: dose of toxic compound required to kill half of the total rats.

**Table 2 molecules-27-07348-t002:** Results of molecular docking.

Name	-CDOCKER Interaction Energy (kcal/mol)	
	wt	Y188CM-RT
NVP	40.9570	38.8888
Lig 1	34.7014	52.3722
Lig 2	26.4688	42.9788
Lig 3	27.6387	44.7056
Lig 4	26.7918	41.0572
Lig 5	40.0960	45.3174
Lig 6	43.7165	45.1867
Lig 7	38.3035	46.1387
Lig 8	38.5926	47.2030
Lig 9	38.5508	48.2571
Lig 10	41.6896	48.7245

**Table 3 molecules-27-07348-t003:** Difference in binding energy with Y188CM-RT and wt.

Type	Samples	Mean ± SD	*p*
wt	10	35.6550 ± 6.4401	<0.05 *
Y188CM-RT	10	46.1941 ± 3.1723	

*p* value < 0.05 was considered statistically significant (Pearson Correlation Coefficient). * Represent significant difference.

**Table 4 molecules-27-07348-t004:** MM/PBSA binding free energies of top 3 ligands with Y188CM-RT.

Entry	ΔG_MM/Van_(kcal/mol)	ΔG_MM/Ele_(kcal/mol)	ΔG_PB_ (kcal/mol)	ΔG_SA_ (kcal/mol)	ΔG_Bind_ (kcal/mol)
Lig 1	−3981.3802	−22,549.8082	−7350.4749	159.4452	−25,725.4551
Lig 9	−4029.5507	−22,448.9056	−7365.6155	160.0775	−25,970.9303
Lig 10	−4024.0403	−22,799.7019	−7020.5051	161.0332	−25,752.7602
NVP	−3987.5892	−22,500.7683	−7328.0362	162.2186	−25,005.7664

ΔG_MM/Van_: Van der Waals energy; ΔG_MM/Ele_: Electrostatic energy; ΔG_PB_: polar solvation, Poisson–Boltzmann term; ΔG_SA_: non-polar solvation −94.7252.

## Data Availability

The datasets generated during the current study are available from the corresponding author on reasonable request.

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
