# Peer review of "Me-Better Drug Design Based on Nevirapine and Mechanism of Molecular Interactions with Y188C Mutant HIV-1 Reverse Transcriptase"

_molecules, 2022, doi:10.3390/molecules27217348_

Round 1

Reviewer 1 Report

The study is a significant research towards optimization of nevirapine. Being an in silico study, there are some more requirements that need to be fulfilled for the completeness of this work.

1. Sufficient data must be provided for ADMET studies in the manuscript.

2. In figure 6, the ligands are superimposed and represent proper binding. However, comparative studies can be incorporated in the same diagram that includes superimposition of ligands along with the reference compound, mentioning the interactions as well.

3. MD simulation data is insufficient. RMSD, RMSF, mmGBSA/mmPBSA are a must.

4. RMSD graph does not show stabilization within performed time slot. It can be extended up to 100 ns. Also, units are not mentioned in the graph axes.

5. In molecular docking and binding energy calculations, three ligands with highest binding energy are wrongly mentioned. Ligands 1, 9 and 10 should be stated instead of ligands 1, 2 and 10.

6. Significance of paired t-test and its relevance to this study needs to be mentioned properly, along with major corrections in the explanation, data, p value and t value. p value is wrongly mentioned in table 2. 

7. Binding affinities are wrongly stated in line 184-185. Kindly correct the same.

8. Appropriate concluding statements can be incorporated for the data represented in table 1.

9. Manuscript lacks a list of abbreviations used.

10. Kindly check for the uniformity of references as per the journal style.

Author Response

We have revised the manuscript in accordance with your comments, and carefully proof-read the manuscript to minimize typographical, grammatical, and bibliographical errors. The amendments are displayed as “Track change” mode in the revised manuscript. Point by point responses are uploaded below.

We would like to thank you again for taking the time to review our revised manuscript.

Reviewer 2 Report

The manuscript describes a computational analysis to modify the drug NVP, to increase the affinity for one of the HIV reverse transcriptase mutants, Y188C.

The authors sufficiently describe the work and conclusions are supported by the obtained results, but some revisions are needed.

Major:

1)    Y188C is a mutant, with Tyr substituted by Cys, not “Cys to Leu” as indicated in line 202.

2)    The docking simulations were performed directly on the mutant structure obtained by homology modelling. No investigation of possible structural modification induced by the substitution was performed. An MD simulation of mutant model can be performed to verify the structural effect, at least around the residue substitution.

3)    From Table 3 NVP does not bind the wt form of reverse transcriptase (binding energy= 112.466 kcal/mol). But it isn’t the drug normally employed for HIV treatment?

The authors must compare the binding energy computationally estimated with experimental values, also obtained by literature. It is well known that docking energy estimation is acceptable as values in a relative way compared to other docking energies, but not as absolute values. Overall, a comparison with experimental data is necessary to evaluate the truthfulness of the results.

Minor:

1)    Explain NVP in the main text, not in the abstract

2)    Check English, see line 104-105 as example.

Author Response

We have revised the manuscript in accordance with your comments, and carefully proof-read the manuscript to minimize typographical, grammatical, and bibliographical errors. The amendments are displayed as “Track change” mode in the revised manuscript. Point by point responses are uploaded below.

We would like to thank you again for taking the time to review our manuscript.

Round 2

Reviewer 2 Report

The authors revised several part of the manuscript, but some revisions  are still needed:

- Please check acronyms must be explained the first time are used in the main text, not in the abstracts. After this, the acronym must be used, not the extended version again.

- "De Novo Evaluation" needs a reference, if it is a known protocol.

- There are still several typos (for ex: "identify of 91%", instead of identity of 91%, "-CDOCKER" repeated more than one time, capitol letters without any reason, etc). Please substitute WT with wt.

- How are ADMET parameters obtained? no detail in the methods section are reported.

- Please use "binding energy" instead of "binding activity". All reported binding energies are negative values? Use a table to report all binding energy values of the LigX for wt and mutant.

- A large English revision is required.

Author Response

Thanks again for your help and advice. We tried our best to revise the manuscript in accordance with your comments, and carefully proof-read the manuscript to minimize typographical and grammatical errors. The amendments are displayed as “Track change” mode in the revised manuscript.

Point by point responses are uploaded below this letter.
